# The Impact of Population Characteristics and Government Budgets on the Sustainability of Public Buildings in Korea's Regional Cities

**Junlae Kim [1] and Seiyong Kim [2],***

[1] National Public Building Center, Architecture & Urban Research Institute, Sejong 30103, Korea; jlkim@auri.re.kr

[2] Department of Architecture, Korea University, Seoul 02841, Korea

\* Correspondence: kksy@korea.ac.kr; Tel.: +82-2-3290-3914

**Abstract:** Public buildings, such as community centers, public libraries, police stations, and fire departments, reflect residents' quality of life. In order to be sustainable, public buildings must reflect regional demographic characteristics and use financial resources effectively. Despite difficulties in tax revenues due to the country's aging and decreasing population, as well as concerns regarding regional extinction, public buildings are being revitalized in Korea's regional cities. Accordingly, this study analyzes the influence of changes in demographic characteristics—particularly in terms of population aging and decline—and regional finances on public buildings in regional cities in Korea. Results show that regions with a lower risk of population decline have a larger area of public buildings, confirming that public buildings reflect changes in population size and the provision of public services. By identifying the impacts of demographic and financial characteristics on public buildings, the findings of this study can facilitate the sustainability of public buildings in regional cities. Based on its findings, this study proposes that regions expand elderly welfare facilities in light of their changing demographic structure. This study's results also underscore the need for the careful consideration of local finances and dependent funding when constructing public buildings.

**Keywords:** public buildings; sustainability; demographic characteristics; population aging; Korea

## 1. Introduction

### 1.1. Crisis of Regional Cities and Public Buildings

The availability of public services significantly influences the selection of residential areas, particularly insofar as satisfaction with these services is directly related to quality of life [1]. As the benefits of most public services are limited to the region, joint consumption results in congestion. Meanwhile, according to public goods theory, it is difficult to find public services without facing issues of exclusivity and rival consumption [2,3]. Therefore, a collaborative relationship between the central and regional governments is essential in ensuring the effective supply and fair distribution of public services. Moreover, with a growing number of countries opting for decentralization, the role of local governments has become increasingly important. Indeed, residents dissatisfied with the regional public services tend to engage in "foot voting"—political competition resulting in regional governments improving available services [4].

While the definition of "public buildings" differs from one country to another, there is consensus that such buildings serve the purpose of providing public services [5]. People's lives are shaped by the benefits of public services. As public services are directly correlated with living standards, the provision of public service facilities is deeply enmeshed in maintaining living environments and

forming communities [6]. In addition to constituting the starting point in the production and supply of public services, public buildings serve as physical spaces and environments. The buildings themselves can serve symbolic purposes, providing regional residents with a sense of satisfaction and pride [7]. However, in analyzing public buildings, what to measure and how—particularly in terms of supply and demand—remains unclear.

According to the World Health Organization (WHO), population aging is a global phenomenon—albeit one unique to each country [8]. Korea's aging population is increasing at an unprecedented rate, an issue compounded by general population decline. These phenomena have emerged as a result of changes in the social structure that have made childbirth and childrearing difficult, as well as increasing life expectancy. More specifically, population decline occurs with the acceleration of population outflow to larger cities, especially young people, which in turn affects regional finances. Some scholars argue that the crises of regional cities began with the excessive administrative and budgetary powers of the central government, with differences in the capacities of regional cities further reinforcing this structural imbalance [9].

According to Article 2 of the Act on the Promotion of Building Service Industry, Korea defines public buildings as "a building or spatial environment built or created by a public institution." While the legal definition of public buildings focuses on the supplier of the building, actual policies and systems emphasize their significance as physical spaces providing public services. In particular, priority is given to improving the design quality of public buildings based on their public nature. Public buildings are typically located in areas favorable to citizens (consumers), and strive to create balance with the surrounding buildings—objectives achieved by ensuring that the business plan is appropriate, using private experts for project oversight, employing public architects, encouraging design competitions and guidelines, and mandating planning work in architecture. The Framework Act on Building—the highest-tier of architectural legislation in Korea—also directs public building development and construction. Recognizing the publicness of living spaces, this Act seeks to secure social publicness and implement cultural publicness. Accordingly, public buildings must be able to accommodate the diverse demands of citizens and social change. However, Korean public building policy fails to consider dynamic changes in regional demographic structures and sizes. Indeed, statistics on public buildings are only collected in 17 cities and provinces, and lack in-depth discussion on methods for systematic operation and management of such buildings, or issues of supply and demand. Certainly, with many of Korea's cities now categorized as "super-aged" and regions showing marked population decline, public building policies require significant change.

Recently, Korea's central government announced a plan for life social overhead capital (SOC) complexation, including developing group public buildings that provide services such as gyms, libraries, and daycare centers. Approximately half of the planned 289 projects have been earmarked for regional cities [10]. In this respect, policies guiding public building expansion and associated central government funding should be welcomed. However, as such expansion will cost more than the sum provided by the central government, thus requiring expenditure from the regional budget, decisions need to be made on the basis of sound research and analysis of demand in regional cities. Meanwhile, given the rapid pace of aging and population decline in Korea's regional cities, this life SOC complexation may lead to the large-scale neglect of public buildings. Addressing these issues, this study analyzes the influence of changes in the demographic characteristics—particularly in terms of population aging and decline—and financial conditions on public buildings in regional cities in Korea, and discusses the implications of these impacts for the sustainability of public buildings. The findings will be useful for the supply and demand management of public buildings in Korea's regional cities, and will have implications for the country and cities where low birth rates and aging population are increasing.

### 1.2. Literature Review and Purpose of Research

This study draws on two research streams: Demographic characteristics and finances, and the supply and demand of public services. Research on demographic characteristics and finances reveals that the reduction of the working-age population and growth of the elderly population can reduce economic participation and labor productivity. In this respect, changes in consumption patterns according to age may influence the revenues of regional governments [11]. More specifically, elderly populations appear to have a negative impact on regional finances because the elderly have less income compared to the working-age population [12]. Scholars also show that the revenue structures of regional governments face different influences from the central government depending on the region, and small internal changes can have significant repercussions [13]. Other studies have shown that the finances of regional cities with larger elderly populations may be further degraded as a result of Korea's taxation structure and method, which focuses on property rather than income and consumption [14–17]. Others have utilized the expenditure decision model to identify the influence of demographic characteristics on regional expenditure, finding that regional expenditure is determined by resident income, level of urbanization, and population density factors [18,19]. In addition to population density, studies were conducted to review the impact of aging on fiscal spending through simulation. A study analyzing the impact of population decline and aging on central and local fiscal spending from 2006 to 2030 predicted that future aging would have a greater impact than population decline on central and local fiscal spending [20]. In addition, an analysis of the impact of an increase in the proportion of the aging population on OECD (Organization for Economic Co-operation and Development) countries showed that fiscal spending related to aging would increase rapidly around 2050 [21]. These findings emphasize that aging can worsen the soundness of local finance and further hamper the supply of public services.

Regarding research on the supply and demand of public services, scholars have utilized demographic, social, economic, and geographic characteristics as variables in order to identify the principle of supply and demand in regional public services. Most studies empirically demonstrate that the supply of public services differs according to specific regional conditions or demographic characteristics. However, perspectives vary depending on the period of research. Early research focused on analyzing the influence of population density and socioeconomic variables on the supply and demand of specific public services, as well as identifying the causes behind regional gaps. In this respect, numerous studies found that the supply of public services increases with higher population density, leading to higher expenditure [22,23]. Scholars interpreted regional gaps as driven by factors relating to race, levels of economic activity, and age groups [24,25]. In particular, a study that revealed that residents living in areas with excellent residential conditions had higher public service satisfaction than those living in underdeveloped areas further emphasizes regional gaps. Both qualitatively and quantitatively, in underdeveloped areas such as regional cities, it is highly likely that even places that provide public services will not satisfy consumers, which could further increase regional decline [26].

Recent studies have focused on the location equity and accessibility of regional public services. Research has revealed the need for multiple optimal locations rather than a geographical center, emphasizing accessibility in the provision of public services, as well as which priorities should influence the location of public services [27–29]. Scholars have also highlighted the relatively low accessibility of public facilities among lower income classes and pedestrians [30], with some suggesting the need for universal design principles to improve the elderly's accessibility to public facilities [31]. Utilizing population density as a common demand factor for public services, studies conducted abroad reveal that sustainability is dependent on fiscal revenues.

There have been a number of empirical studies on changes of demographic characteristics in Korea. However, unlike those conducted abroad, these studies focus on population decline, which is regarded as a major social issue and the cause of a national crisis in Korea. Scholars typically explain the outcomes of demographic changes in terms of population size and structure, which have positive or negative influences on regional communities [32]. More specifically, changes in the demographic

structure lead to changes in the demand for public services; population declines lead to a reduced supply of public service facilities, impacting living standards. These scholars argue that the reduction in tax revenues results in an inability to maintain public service facilities, leading to a decline in the quality of public services due to the poor maintenance and repair of such facilities [33,34].

Based on existing research, this study analyzes the influence of changes in demographic characteristics—namely, aging and population decline—and financial conditions on public buildings in regional cities in Korea. To achieve this, this study constructs panel data integrating time series and cross-sectional data for empirical analysis in order to closely examine the dynamic changes in regional cities in Korea. In doing so, this study contributes to policies regarding the supply and demand of public buildings in regional cities in light of intensifying population decline.

## 2. Changes in the Demographic Characteristics of Korea's Regional Cities and the State of Public Buildings

### 2.1. Changes in Population Structure and Population Size

In 2000, more than 7% of Korea's population was above the age of 65, making the country an aging society. In 2017, the proportion of the elderly population passed 14%, categorizing Korea as an aged society [35]. Today, Korea is on the precipice of becoming a super-aged society, with 20% of its population expected to be over the age of 65 by 2026. This speed of aging is unprecedented—Korea experiences a rate double that of developed countries in the past. Meanwhile, Korea's total fertility rate has continued to decline; after reaching 1.24 babies in 2015, fertility dropped to 0.98 in 2018, and 0.92 babies in 2019 [36]. This is the lowest figure among the OECD countries, with Korea the only country to report a fertility rate below 1 for two years in a row. Based on changing fertility rates, the Korean population is expected to start decreasing as early as 2019 [37].

Natural population growth, which reached 210,000 in 2011, fell to 8000 in 2019. The natural decline of the population—where deaths outnumber births—is expected to begin in 2020, with the working-age population predicted to drop by 2.5 million in 2028 compared to 2018 (Figure 1). A more significant problem is that regional cities are more vulnerable to aging and population decline compared to metropolitan areas and metropolises. With the outflow of the working-age population increasing, this phenomenon is compounding a vicious cycle of deteriorating regional finances.

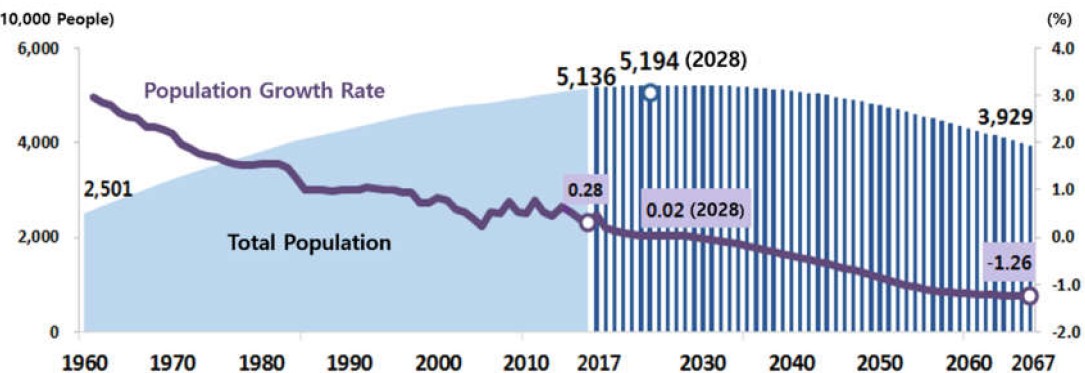

**Figure 1.** Forecasts of Korea's total population and population growth rates. Data: Special estimate of the future population: 2017–2067.

Figure 2 shows the aging trends in 121 regional cities over the study period (2012–2018). While the regions with an aged society—that is, regions were more than 14% of the population is elderly—have declined from 26 to 24, super-aged regions, where 20% or more of the population is elderly, have increased from 70 to 85. Aging intensified in all regional cities between 2012 and 2018. Currently, regional cities classified as super-aged have already surpassed 70% of the total and

are experiencing weakening regional competitiveness in the form of lower labor productivity and weakened industry competitiveness.

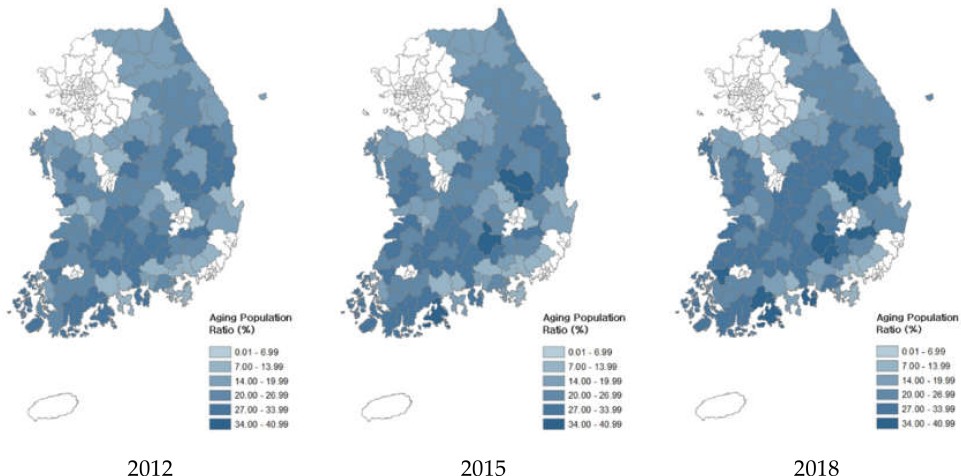

**Figure 2.** Changes in the elderly population ratio (population structure) in regional cities.

Based on Masuda's explanation of "regional extinction" [38], which indicates that the outflow of the young female population can lead to the extinction of a region, Lee's extinction risk index compares the elderly population aged 65 and over and the female population aged between 20 and 39 [39]. If the ratio of the elderly population presented earlier results in changes to the population structure with an increase in the proportion of the elderly in the entire population, then the regional extinction risk index, based on the outflow of the young female population, indicates the reduction of regional populations.

Figure 3 illustrates the change in the extinction risk index of regional cities. If the extinction risk index is less than 0.5, a region is considered to be entering the extinction risk stage; if it is less than 0.2, a region is considered a high-risk extinction area and classified as an area with a high extinction risk. Figure 3 shows that the number of regions entering the extinction risk stage increased from 70 in 2012 to 71 in 2018. Moreover, while the regions in the high-risk extinction area did not exist in 2012, 13 such regions had emerged by 2018. Currently, more than 77% of Korea's regional cities are classified as in the extinction risk stage. As such, Figure 3 indicates that regional extinction may actually occur following population decline.

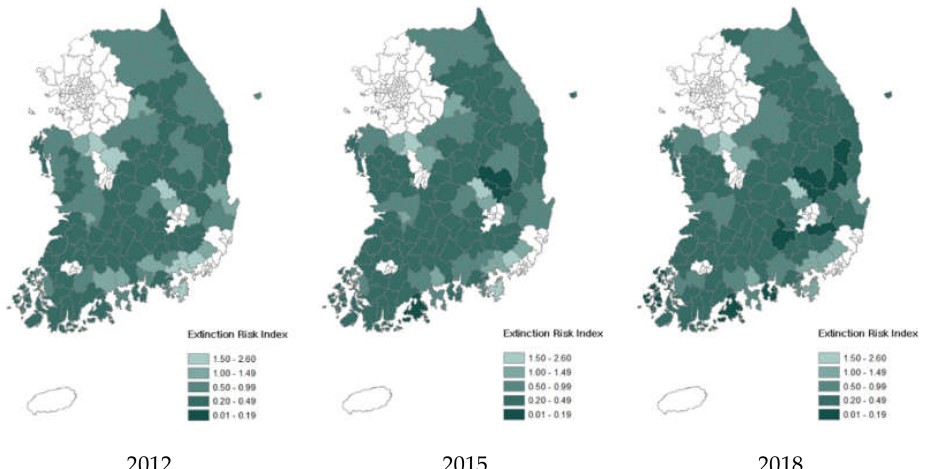

**Figure 3.** Extinction risk index (population decline).

In summary, regional cities in Korea are experiencing intensifying population decline, as well as rapid aging and low fertility rates, due to the outflow of young people. Significantly, these changes are

rapid and not limited to certain regional cities. Changing trends in demographic characteristics in regional cities also indicate that the emergence of natural population decline threatens Korean stability.

### 2.2. The Status of Public Buildings in Regional Cities

As of 2018, Korea had a total of 210,441 public buildings, with a total area of 209,656,000 m$^2$. Public buildings are increasing at a rate of 5000 buildings per year. As Figure 4 shows, the average annual rate of increase of public buildings is 2.5%—approximately 2.5 times higher than that of buildings across the whole country (1.0%) [40]. In other words, the number of public buildings has continued to increase, representing 3% of all buildings in Korea and 5% of the total building area. In terms of total area of public buildings by use, educational and research facilities—such as schools and libraries—represent the largest portion at 86,003 thousand m$^2$ (41.0%), followed by public work facilities such as government buildings at 23,134 m$^2$ (11.0%). Public building use is concentrated in these two facilities.

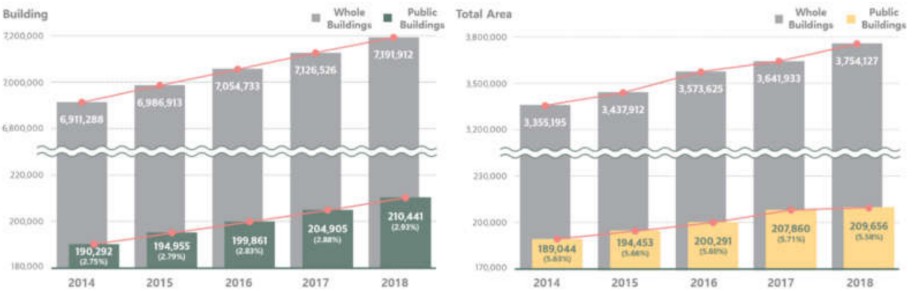

**Figure 4.** Comparison of the current status of national and public buildings. Data: Public Building Statistics 2018 (NPBC).

The areas of public buildings providing public services have continued to increase. Figure 5 shows the changes in the total area of public buildings in regional cities over the research period. The number of regions with more than 500,000 m$^2$ increased from 43 in 2012 to 63 in 2018, while the number of regions with more than 1,000,000 m$^2$ increased from 16 to 22.

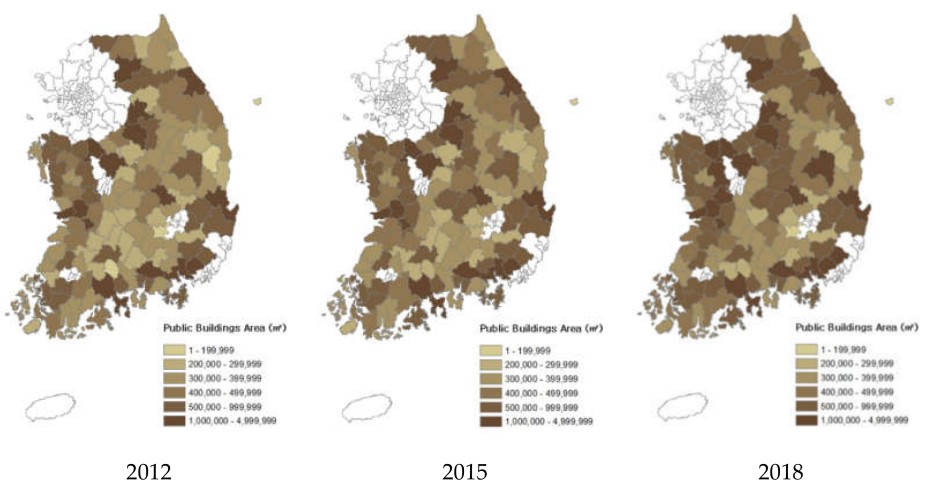

**Figure 5.** Total area of public buildings, 2012–2018.

Public buildings responding to population aging comprise elderly welfare facilities (e.g., elderly homes and long-term care homes), classified as facilities designed for the elderly and children. Population aging makes it necessary to secure elderly welfare facilities providing a diverse range of services within residential zones, as well as budgeting allocation and demand analyses to secure the diversity of these facilities [41]. Figure 6 illustrates the change in the total area of the welfare facilities

for the elderly as a portion of public buildings in regional cities during the study period. Regions with a total area of more than 10,000 m$^2$ increased from 38 in 2012 to 50 in 2018, while regions with more than 20,000 m$^2$ expanded from 10 to 16.

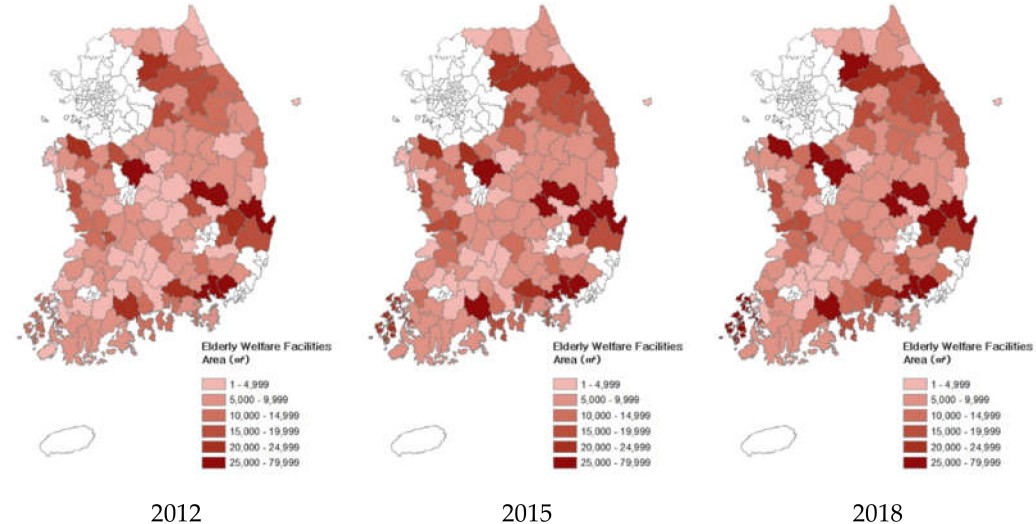

2012　　　　　　　　　　　　　2015　　　　　　　　　　　　　2018

**Figure 6.** Changes in total area of elderly welfare facilities among public buildings in regional cities, 2012–2018.

In addition to the increase of public buildings, societies need to consider their sustainability in light of natural population decline. Figure 7 shows the changes in the service levels of public buildings per 1000 people in metropolitan areas, revealing that buildings increased from 1.81 in 2012 to 1.98 in 2018, while the total area increased from 2.84 m$^2$ to 3.32 m$^2$. In the same period, public buildings in regional cities increased by 1.18 buildings—an increase six times that of metropolitan areas. Meanwhile, the total area increased from 4.91 m$^2$ to 5.82 m$^2$, twice that of metropolitan areas. This indicates a very high level of public buildings provided per unit of the population in Korea's regional cities.

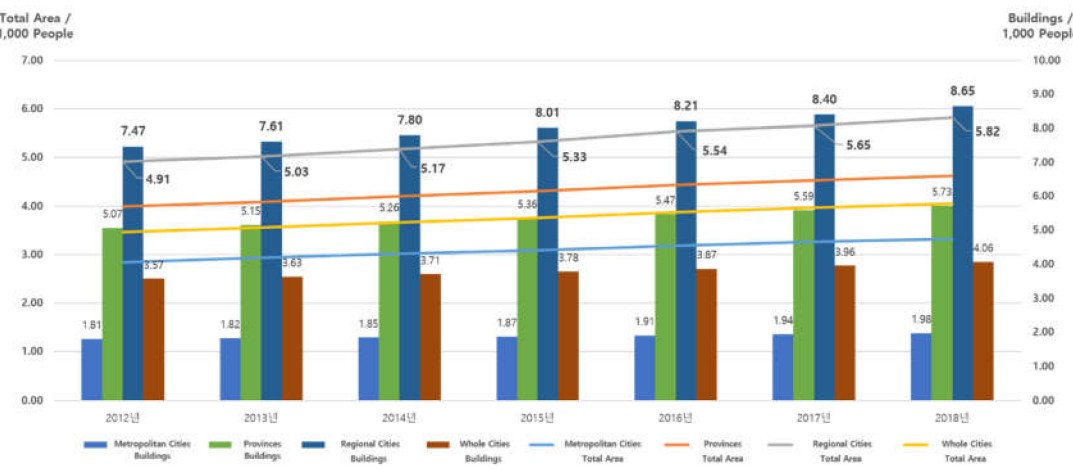

**Figure 7.** Comparison of public building provision levels per 1000 people by region, 2012–2018.

As such, the supply of public buildings in regional cities does not appear to have responded appropriately to changes in population characteristics. Given issues of an increasing elderly population and regional extinction due to population decline, increases in public buildings should be viewed critically. Therefore, this study analyzes the influence of changes in the population structure and size of regional cities, as well as the financial impact of population decline on public buildings in regional cities.

### 3. Study Design and Methodology

#### 3.1. Analytical Approach

Korea is comprised of eight metropolitan cities and nine provinces. This study analyzes regional cities located in seven provinces, covering 121 local governments (cities and counties). The study area was limited for two reasons. First, changes to the population structure as a result of aging and population decline are not severe in metropolitan areas. Second, the statuses of regional finances are relatively sound, presenting a high degree of freedom in policies relating to public buildings. Among the nine provinces, Gyeonggi-do and Jeju Special Self-Governing Province were excluded from the analysis because Gyeonggi-do is located in a metropolitan area, and Jeju Island is separated from the mainland. Moreover, Jeju Island may be difficult to classify as a province under the Local Autonomy Act and the Special Act on the Establishment of Jeju Special Self-Governing Province. Therefore, the 121 regional cities analyzed by this study are regions facing changes to their population structure, including aging and intensifying population decline due to the outflow of the working-age population and low fertility rates, and thus facing financial difficulties.

This study selected the demographic factors and regional finance factors used in the extant research. In terms of the demographic factors relating to the consumers of public services, this study constructed variables in relation to population structure (aging) and population decline (extinction risk index). This study divides regional finances into revenue and expenditure budgets and analyzes the influence of the source of funding and funding area on public buildings. As a result, 121 regional cities were required to be analyzed cross-sectionally while reviewing their time series by year. Therefore, the multi-panel regression methodology was used. To examine the dynamic changes, this study constructed balanced panel data comprising both cross-sectional and time-series data on regional cities. As a result of data acquisition conditions, the study period was set at 2012–2018.

#### 3.2. Data and Variable Composition

This study's dependent variable is the total area of public buildings in regional cities. Although various criteria could have been considered in addition to the total area, it was judged that the total area was most appropriate in the area of objectification and quantification due to the cross-section and time-series analysis of 121 regional cities. As the area of the individual public building is calculated using the minimum standards and the degree of demand for the public services provided, the level of public services provided needs to be measured. The buildings can be compared under the same conditions irrespective of their designated purpose. This study added another dependent variable—namely, the public buildings designated as elderly welfare facilities—in order to analyze the response to aging by regional cities. For analysis, the total area is used in the same way as public buildings. The data on the total area of public buildings were sourced from the public building data owned by the central and local governments, which are registered in the electronic architectural administration information system (Seumteo). For this study, the original data of public buildings were selected from the Seumteo database, so reliability can be assured. Data were collected at the end of December for each year of the study period in order to construct the panel data. In cases of changes to administrative areas, such as consolidation or promotion from a county to a city, during the study period, data were modified and supplemented in 2018.

The demographic factors comprising the independent variables include the elderly population ratio, extinction risk index, and total fertility rate. The extinction risk index was selected as a proxy for population decline and formulated in a separate model to compare its influence with that of population structure as impacted by the elderly population ratio. The regional finance factors were classified into revenue budgets by source and expenditure budget by function. Revenue budget by source identifies the main sources of a regional city's revenue, including regional tax and non-tax revenues, as well as local share tax and treasury grants, which function as the local finance coordination system.

Furthermore, while it is not directly classified in the revenue budget, this study also considers the financial autonomy variable in order to account for the autonomy of regional cities in fiscal management.

Major functions of the expenditure budget are determined based on the purpose of expense execution. Among the 14 functional classifications of expenditure budgets, this study utilizes the variables of general public administration, public order and safety, education, culture and tourism, social welfare, health, agriculture, and marine and fisheries based on their relationship with public buildings, as well as omissions of budgets by function. The demographic factors in the independent variables were sourced from official statistics. The extinction risk index, which is not provided in the official statistics, was calculated on an annual basis for 121 local cities by processing resident registration population using the analysis methodology of leading researchers Masuda and Lee Sang-ho. Other factors relating to regional finances were sourced from the Local Finance Integrated Open System; they were subjected to the same standard as the dependent variables, and measured at the end of December each year. However, it is necessary to exclude variables with correlation coefficients of 0.7 or higher before conducting panel regression analysis. As such, regional tax and treasury grants were removed from tax revenues, and social welfare was removed from expenditure budgets in the final analysis model (Table 1). Four models were used for the analysis. Model 1 used the aging population ratio and the revenue budget variables to examine the population structure and the impact of the revenue budget, while Model 2 used the extinction risk index instead of the aging population ratio to review the population size and revenue budget. On the same principle, Model 3 and 4 formed a model by substituting expenditure budgets instead of revenue budgets. This study then conducted multiple panel regression analyses based on the following models:

**Table 1.** Data description.

| Variables | | | Description |
|---|---|---|---|
| Dependent Variable | | Public Buildings Area | Total floor area of public buildings |
| | | Elderly Welfare Facilities Area | Total floor area of elderly welfare facilities among public buildings |
| Independent Variables | Population characteristics factors | Elderly Population Ratio | Population 65 or older ÷ Population × 100 |
| | | Extinction Risk Index | Female population (aged 20–39) ÷ Elderly population (aged 65 and above) |
| | | Total Fertility Rate | Sum of fertility rates by age (15–49 years) ÷ 1000 |
| | Local finance factors / Revenue budget | Financial Autonomy | (Own income + own funding source) ÷ Local government budget size × 100 |
| | | Non-Tax Revenue | Property rental income, utilization fee income, fee income, etc. |
| | | Local Share Tax | Transfer of funds from the state or higher-level local government to provide efficiency in fiscal operations |
| | Expenditure budget | General Public Service | Total amount for legislative and election management, regional administration, and financial support, as well as fiscal, financial, and general administration areas |
| | | Public Order & Safety | Total amount for police, disaster prevention, civil defense, and firefighting |
| | | Education | Total amount for early childhood, primary and middle school education, higher education, and lifelong job education |
| | | Culture & Tourism | Total amount for culture and arts, tourism, sports, cultural properties, and general culture and tourism |
| | | Health | Total amount in healthcare and food and drug safety |
| | | Agriculture, Forestry, and Fisheries | Total amount in agriculture, farming, forestry, marine and fisheries, and fishing villages |

Model 1: Population structure (elderly population ratio) and revenue budget

$$\ln PBA(EWFA) = \alpha + \beta_1(EPR_{rt}) + \beta_2(TFR_{rt}) + \beta_3(FA_{rt}) + \beta_4(\ln NTR_{rt}) + \beta_5(\ln LST_{rt}) + u_r + \varepsilon_{rt} \quad (1)$$

Model 2: Population size (extinction risk index) and revenue budget

$$\ln PBA(EWFA) = \alpha + \beta_1(ERI_{rt}) + \beta_2(TFR_{rt}) + \beta_3(FA_{rt}) + \beta_4(\ln NTR_{rt}) + \beta_5(\ln LST_{rt}) + u_r + \varepsilon_{rt} \quad (2)$$

Model 3: Population structure (elderly population ratio) and expenditure budget

$$\ln PBA(EWFA) = \alpha + \beta_1(EPR_{rt}) + \beta_2(TFR_{rt}) + \beta_3(\ln GPS_{rt}) + \beta_4(\ln POS_{rt}) + \beta_5(\ln ED_{rt}) + \beta_6(CT_{rt}) \quad (3)$$

Model 4: Population structure (extinction risk index) and expenditure budget

$$\ln PBA(EWFA) = \alpha + \beta_1(ERI_{rt}) + \beta_2(TFR_{rt}) + \beta_3(\ln GPS_{rt}) + \beta_4(\ln POS_{rt}) + \beta_5(\ln ED_{rt}) + \beta_6(CT_{rt}) \quad (4)$$

In these models 1–4, ※ r refers to regional city, t refers to year (2012–2018), u refers to an error term indicating unobservable characteristics, and $\varepsilon$ is an error term.

## 4. Results and Discussion

### 4.1. Verification of Panel Data

First, the explanatory power of the four types of panel models ranged between 76.6% and 79.7% for public buildings, and 32.3% and 39.4% for elderly welfare facilities. However, this study conducted time-series autocorrelation and heteroscedasticity tests for model 1–4 to test the assumptions of the panel data, and found the existence of both autocorrelation and heteroscedasticity (Table 2). Accordingly, this study utilized FGLS(Feasible Generalized Least Squares) to resolve issues of autocorrelation and heteroscedasticity and reconducted the analysis.

**Table 2.** Verification of demographic changes and financial structure analyses.

| | Model (1) | | Model (2) | | Model (3) | | Model (4) | |
|---|---|---|---|---|---|---|---|---|
| | PBA | EWFA | PBA | EWFA | PBA | EWFA | PBA | EWFA |
| $R^2$ | 0.7842 *** | 0.3226 *** | 0.7970 *** | 0.3601 *** | 0.7887 *** | 0.3902 *** | 0.7660 *** | 0.3944 *** |
| Autocorrelation | 100.218 *** | 224.225 *** | 115.607 *** | 236.208 *** | 107.182 *** | 267.093 *** | 138.983 *** | 326.579 *** |
| Heteroskedas ticity | 29,813.85 *** | 96,532.25 *** | 19,616.16 *** | 49,862.71 *** | 34,368.16 *** | 1.40e+05 *** | 22,718.09 *** | 33,612.86 *** |

* $p < 0.1$, ** $p < 0.05$, *** $p < 0.01$.

### 4.2. Interpretation of Analysis Results

Table 3 shows the results of this study's analysis of changes in demographic factors and revenue budgets. In order to classify the degree of influence of population structure and population decline, Model 1 utilized the independent variable of the elderly population ratio, while Model 2 utilized the extinction risk index as a proxy for population decline. The individual models show the results of FGLS following assumptions based on the previous verification results; the Wald-chi$^2$ values are all statistically valid at a significance level of 0.01. Model 1 shows that the demographic factors of the elderly population ratio and total fertility rate negatively influence public buildings. In terms of revenue budget, financial autonomy and non-tax revenues had a negative influence, whereas local share tax had a positive influence. Model 2 shows that the extinction risk index positively influences public buildings, while total fertility rate has a negative influence. Meanwhile, in terms of revenue budgets, only local share tax positively influences public buildings. However, in both Models 1 and 2, total fertility rate was not statistically significant for elderly welfare facilities, and the significance level of financial autonomy was comparatively low.

**Table 3.** Analysis of the demographic characteristics-revenue budget model.

| Variables | Model (1) | | | | Variables | Model (2) | | | |
| | PBA | | EWFA | | | PBA | | EWFA | |
| | Coef. | z | Coef. | z | | Coef. | z | Coef. | z |
|---|---|---|---|---|---|---|---|---|---|
| Elderly Population Ratio | −0.0535 *** | −40.53 | −0.0341 *** | −13.90 | Extinction Risk Index | 1.1718 *** | 38.7 | 0.7158 *** | 12.97 |
| Total Fertility Rate | −0.0563 *** | −3.25 | −0.0280 | −1.03 | Total Fertility Rate | −0.0722 *** | −4.16 | −0.0326 | −1.18 |
| Financial Autonomy | −0.0033 *** | −3.36 | −0.0031 * | −1.83 | Financial Autonomy | −0.0051 *** | −5.49 | −0.0039 ** | −2.37 |
| Non-Tax Revenue | −0.0292 *** | −7.79 | −0.0200 *** | −3.34 | Non-Tax Revenue | −0.0249 *** | −6.66 | −0.2129 *** | −3.52 |
| Local Share Tax | 0.6194 *** | 29.68 | 0.4807 *** | 14.13 | Local Share Tax | 0.6097 *** | 30.05 | 0.4590 *** | 13.43 |
| Constant | 7.4428 *** | 27.86 | 4.4334 *** | 10.25 | Constant | 5.8004 *** | 22.15 | 3.6093 *** | 8.17 |
| Wald-chi * | 2383.11 *** | | 342.07 *** | | Wald-chi * | 2171.84 *** | | 323.64 *** | |

$* p < 0.1, ** p < 0.05, *** p < 0.01$.

As such, public buildings in regional cities are more likely to increase with lower elderly population ratios, lower risk of population decline, and higher budget dependence on the central government and larger local governments. Furthermore, the fact that the regression coefficient of the extinction risk index is the highest indicates that public buildings have been supplied to regions with low risk of population decline. Two aspects need to be noted. First, the negative influence of the elderly population ratio on elderly welfare facilities indicates that these buildings do not reflect the trend of population aging. In order to reflect these changes, population structure changes must be accepted by expanding publicly operated and managed elderly welfare facilities. Second, the regression coefficient of local share tax is very high, indicating that regional cities can expand public buildings using their dependent funding. A minimum level of public services needs to be provided, even in regions experiencing population decline, and expanding elderly welfare facilities is desirable given the aging of the population. However, caution is necessary given the potential wasting of significant local share tax funding at the national level, resulting in the neglect of luxury government buildings and new public buildings. Therefore, a careful approach toward the uses and functions of public buildings is necessary.

Table 4 shows the analysis results of Model 3, which considers the changes in population structure, and Model 4, which considers population decline. As expenditure budgets are classified based on the purpose of budget execution and their functions, it is possible to indirectly estimate the characteristics of public buildings based on the influence of variables. First, the FGLS analysis and Wald-chi$^2$ test revealed that the individual models were statistically valid. Model 3 shows that all expenditure budgets except that of public order and safety had a positive influence on public buildings. In contrast, the demographic factors of the elderly population ratio and the total fertility rates had a negative influence. Model 4 also shows that all expenditure budgets except that of public order and safety positively influenced public buildings. While the extinction risk index had a positive impact on public buildings, the total fertility rate had a negative influence.

As such, public buildings in regional cities face little risk from the elderly population ratio and population decline, and are likely to increase with larger expenditure budgets by function. However, the sustainability of public buildings may be threatened, as more than 70% of regional cities are now classified as super-aged societies, while 77% of regions are classified as being at risk of extinction. Therefore, a public building policy taking the changes in demographic characteristics of regional cities into account is necessary. Furthermore, the regression coefficient of the education variable was the lowest among the statistically significant variables in both models 3 and 4. Considering that the total area of education and research facilities accounts for 41% of the public buildings, as well as the proportion of regional cities classified as super-aged societies, it is necessary to expand policies and facilities for lifelong education.

**Table 4.** Analysis of the demographic characteristics-expenditure budget model.

| Variables | Model (3) | | | | Variables | Model (4) | | | |
|---|---|---|---|---|---|---|---|---|---|
| | PBA | | EWFA | | | PBA | | EWFA | |
| | Coef. | z | Coef. | z | | Coef. | z | Coef. | z |
| Elderly Population Ratio | −0.0459 *** | −35.35 | −0.0266 *** | −10.84 | Extinction Risk Index | 0.9328 *** | 28.03 | 0.5685 *** | 11.78 |
| Total Fertility Rate | −0.1032 *** | −6.00 | −0.0985 *** | −3.82 | Total Fertility Rate | −0.0977 *** | −5.54 | −0.1077 *** | −4.90 |
| General Public Service | 0.0861 *** | 7.91 | 0.0763 *** | 5.43 | General Public Service | 0.0742 *** | 6.88 | 0.0727 *** | 5.89 |
| Public Order & Safety | 0.0020 | 0.43 | −0.0009 | −0.12 | Public Order& Safety | 0.0052 | 1.12 | 0.0011 | 0.15 |
| Education | 0.0194 *** | 5.48 | 0.0181 *** | 3.01 | Education | 0.0215 *** | 5.40 | 0.0243 *** | 4.18 |
| Culture & Tourism | 0.0922 *** | 8.04 | 0.0784 *** | 4.43 | Culture & Tourism | 0.0905 *** | 7.78 | 0.0665 *** | 4.04 |
| Health | 0.1647 *** | 12.34 | 0.1349 *** | 6.91 | Health | 0.1371 *** | 10.05 | 0.1218 *** | 7.10 |
| Agriculture & Forestry & Fishery | 0.2069 *** | 12.07 | 0.2020 *** | 7.34 | Agriculture & Forestry & Fishery | 0.1687 *** | 10.17 | 0.1939 *** | 7.57 |
| Constant | 8.5595 *** | 41.01 | 4.613 *** | 13.64 | Constant | 7.7569 *** | 38.94 | 4.0068 *** | 12.21 |
| Wald-chi* | 3175.63 *** | | 486.35 *** | | Wald-chi* | 2068.79 *** | | 549.66 *** | |

* $p < 0.1$, ** $p < 0.05$, *** $p < 0.01$.

Ultimately, this study confirms that the total areas of public buildings and elderly welfare facilities in regional cities in Korea are more significantly influenced by population size than changes in population structure. However, the fact that the elderly population ratio in regional cities negatively impacts the total area of both public buildings and elderly welfare facilities indicates the need to improve the public building policies of regional cities. Given the increase of regional cities at risk of extinction due to population decline, it is necessary to develop policies for public buildings that provide a minimal level of public services and respond to population aging.

## 5. Discussion and Conclusions

Public buildings are becoming increasingly important as they provide residents with necessary support and benefits and indicate residents' quality of life. Regional cities in Korea are facing the risk of regional extinction as a result of intensifying changes to population structure and population decline, particularly in terms of population aging. Although the deterioration of regional finances has a negative influence on the sustainability of public services, public buildings in regional cities are on the rise. Accordingly, this study sought to analyze the influence of changes in demographic factors and finances on public buildings in Korea's regional cities and identify the implications of these impacts for public building policies. In order to achieve this, this study constructed balanced panel data on the total area of public buildings and elderly welfare facilities for 121 local governments representing regional cities between 2012 and 2018, and then conducted multiple regression analyses. More specifically, this study divided the demographic characteristics into population structure and population size, and regional finances into revenue and expenditures, yielding four models to consider the impact of each factor.

This study's findings can be summarized as follows. First, the total area of public buildings and elderly welfare facilities are more affected by changes in population size arising from population decline than structural changes resulting from population aging. As public buildings have primarily increased around regions with a low risk of population decline, the results of this study confirm that changes in regional demand are reflected by public buildings. However, as more than 70% of Korea's regional cities are considered super-aged societies, the negative influence of the elderly population ratio on the total area of public buildings and elderly welfare facilities indicates the need for improvements in public building policies in regional cities. It is necessary to expand elderly welfare facilities in phases, as well as consider the elderly population in the region. Second, while total fertility rate indicates the beginning of natural population growth due to increasing births, it negatively influences public buildings in regional cities. These results indicate the need to consider childbirth when constructing public buildings. Public building policies should either support and encourage childbirth or minimize the outflow of households with children. Third, the number of public buildings increased with a larger allocation of regional budgets, which rely on the central or regional local governments. This indicates

that failure to consider changes in the demographic characteristics of regional cities may result in budget waste, leading to the neglect of public buildings.

Therefore, the following alternatives are proposed: First, areas where the elderly population ratio and extinction risk index of local cities are higher than average or where public buildings are increasing more rapidly than before need to be monitored at the national level. Nationwide, this is expected to save local grant tax and, at the same time, prevent the deterioration of local finance. Second, in order to cope with changes in population characteristics in the future, it is necessary to prioritize the multi-purpose use of public buildings or the remodeling of public buildings. To minimize budget waste and the neglect of public buildings, the Life SOC complexation plan implemented by the current government requires approaches that comprehensively consider the elderly population ratio, trends in population decline, and characteristics of funding, rather than the application of standardized criteria.

Leading studies revealed that the higher the population size, the greater the supply of public services. In particular, the results of the simulation study, which shows that future aging may worsen local finances more than population decline, further underlines the importance of public service supply, taking into account the demographic characteristics of regional cities as a result of this study. It should be noted that the creation of public buildings in areas where there is no demand can accelerate the decline of the relevant city. In addition, the results of a prior study, which warned that the elderly population could worsen local finances, were also valid for regional cities where population reduction occurs. These findings are significant for countries where low birth rates and aging are beginning, and for regional cities where population outflow is deepening. Therefore, regions with higher elderly population ratios and intensifying population decline should pay closer attention to constructing public buildings through the efficient use of limited regional finances and dependent funding.

While this study expands discussions of regional extinction in Korea's regional cities in light of intensifying issues of aging and population decline, it has some limitations. Notably, there are constraints to interpreting the results of this study, particularly given limitations of acquiring usable data at the level of local government. Further analysis of infant- and child-related uses will help reduce the low birth rate in regional cities, while analyzing the characteristics of sports facilities and community distribution at smaller village levels will provide a close look at the quality of life at the local level. Therefore, future multi-dimensional research requires confirming and utilizing sociodemographic data by regional city, data on the status of free space for public buildings, and detailed data regarding use and functions.

**Author Contributions:** Conceptualization, J.K.; methodology, J.K. and S.K.; software, J.K.; validation, J.K. and S.K.; formal analysis, J.K. and S.K.; visualization, J.K.; supervision, S.K. All authors have read and agreed to the published version of the manuscript.

**Funding:** This research received no external funding.

**Acknowledgments:** This research was supported by Architecture & Urban Research Institute in the use of basic data. The authors wish to express their gratitude for the support.

**Conflicts of Interest:** The authors declare no conflict of interest.

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
