# Peer review of "The Impact of Population Characteristics and Government Budgets on the Sustainability of Public Buildings in Korea’s Regional Cities"

_sustainability, doi:10.3390/su12145705_

Round 1

Reviewer 1 Report

This study analyzed the influence of changes in demographic characteristics and regional finances on public buildings in regional cities in Korea. The authors provided a detailed introduction for readers to understand the research background.

For study design and methodology, some queries should be clarified. First, apart from the total area of public buildings in regional cities, another dependent variable was the public buildings designated as elderly welfare facilities in Section 3.2. Was the public buildings designated as elderly welfare facilities the total area or the number of the buildings? Second, how the data of this study were collected? The author only mentioned the date of data collection but did not give details about the way of collecting data and ensuring the data were reliable. Third, how was the extinction risk index calculated? Fourth, more explanations about the four models should be made in Section 3.2. Fifth, the rationale for using multiple panel regression should be provided in Section 3.2.

For the discussion part, the authors only interpreted the analysis results but failed to indicate the theoretical contributions derived from this study. The authors should discuss the results from the theoretical perspectives. Also, a separate section should be made to focus on the practical implications of this study.

Reviewer 2 Report

This paper addresses an interesting topic on the influence of changes in demographic characteristics and regional finances on the development of public buildings. A case study in regional cities in Korea was provided. The paper fits within the scope of the journal and the methodology is appropriate. Please consider improve the followings:

The problem, research gaps and limitations should be emphasized.

The contribution of this study should be highlighted in the introduction.

In line 213, it is suggested to add some figures and description to introduce the configuration of public buildings. For example, the typical type of flat with size and accommodated people, typical layout plan, etc.

In Chapter 4, while this study focuses on Korea context, will it be possible to generalize the results into different counties. For example, correlate the results with specific type of city configurations, finance status, etc. And discuss the possible uncertainty.

The conclusion is too lengthy, and it is hard to find some new elements in the conclusion. It should be revised with adding more theoretical findings, revelations, and contributions to provide a boarder impact.

Reviewer 3 Report

The manuscript reports on an interesting study seeking to analyze the influence of changes in the demographic characteristics and financial conditions on public buildings in Korean regional cities whilst discussing the sustainability of public building provision therein. This study is considered timely in the face of dwindling government revenue and the need to ensure that provision of public buildings in the right quantity. The following observations have been made for the authors consideration to boost the study's contribution to scholarship. 

  1. The introduction literature review sections are largely fixated on the Korean situation without regard to the incidence of the phenomenon being investigated within the global context. This makes a comparison of the Korean situation to similar or dissimilar situations across other contexts difficult. This is evident in the narrow nature of the discussion section wherein the authors concentrated on presenting the data from the study without a comparison to findings from similar studies. Doing so will assist the authors in drawing easily generalizable conclusions.
  2. Although the literature reviewed is considered barely adequate for the study, it is suggested that, in line with the previous comment, the authors should source and incorporate more literature, especially along the lines of demographic characteristics and finances and, the supply and demand of public services.
  3. The research methodology is well articulated. The variables (dependent and independent) used in the models were explained succinctly to engender comprehension. Also, a robust justification was given for the selection of cases (the provinces within which the phenomenon was investigated. The use of the panel data as well as the source of the data and the procedures deployed to elicit the data sets were highlighted in the manuscript.
  4. The authors should consider the incorporation of a distinct discussion section beyond the presentation of results section within which the study’s results will be discussed in the light of findings from similar studies in different contexts.
  5. Also, the authors should consider deliberating more on the implications of the study’s results beyond the points raised between lines 374 and 375 on page 12 of 15.

Reviewer 4 Report

Substantive comments:

The article “The Impact of Population Characteristics and Government Budgets on the Sustainability of Public  Buildings in Korea’s Regional Cities” analyzed the influence of changes in demographic characteristics—particularly in terms of population aging and decline—and regional finances on public buildings in regional cities in Korea and discusses the implications of these impacts for the sustainability of public buildings.

The subject of the paper is adequate to the concerns of journal with appropriate structure. The article is well structured and overall well written.

At first authors claimed definition and situation of public buildings in Korea and show where is the research problem. They refer to contemporary literature, acts, plans and projects, etc. There is adequate literature review in this topic.

The research methodology is clear and explains quite clearly the limitations related to the conducted research:

The study area 234 was limited for two reasons. First, changes to the population structure due as a result of aging and 235 population decline are not severe in metropolitan areas. Second, the statuses of regional finances are 236 relatively sound, presenting a high degree of freedom in policies relating to public buildings.

Explanation requires what from the point of view of spatial issues for research was taken into account, why only the total area of ​​public buildings in regional cities. Does this variable only affect the programming and designing of public services in the context of demographic change on the regional level? This is not enough when it comes to quality of life.

It is also doubtful whether the buildings can be compared under the same conditions irrespective of their designated purpose. A different approach is required for the swimming pool, which is a sports facility, but also a public facilities, and must meet the relevant standards in terms of space, and a supermarket with a diverse number of stores and warehouses.

Finally, the article promises some recommendations for regional politics, but these are not provided and certainly not in sufficient detail. Such recommendations could also include specific suggestions for example in result: to minimize budget waste and the neglect of public  buildings, the Life SOC complexation plan implemented by the current government requires  approaches that comprehensively consider the elderly population ratio, trends in population decline, and characteristics of funding, rather than the application of standardized criteria.
Including demographic issues in the programming of public service investments at the regional level does not seem to be a satisfactory and in-depth research conclusion.

Conclusions clearly states limitations and potential future studies, but does not elaborate on the implications of the findings.

Technical comments:

Descriptions too small especially in maps, not illegible.

Round 2

Reviewer 1 Report

The authors did a good job of addressing my comments. I have no further comments.

Reviewer 2 Report

The authors have addressed the problems arise. The manuscript has been much improved.

Reviewer 3 Report

The authors have sufficiently addressed the comments made in the previous review. 

Reviewer 4 Report

I am satisfied with the replies.